# Novel 2-Thiouracil-5-Sulfonamide Derivatives: Design, Synthesis, Molecular Docking, and Biological Evaluation as Antioxidants with 15-LOX Inhibition

**DOI:** 10.3390/molecules28041925

**Published:** 2023-02-17

**Authors:** Naglaa M. Ahmed, Ahmed H. Lotfallah, Mohamed S. Gaballah, Samir M. Awad, Moustafa K. Soltan

**Affiliations:** 1Pharmaceutical Organic Chemistry Department, Faculty of Pharmacy, Helwan University, Ein Helwan, Cairo 11795, Egypt; 2Department of Pharmaceutical Chemistry, Faculty of Pharmacy, Sinai University, El-Arish 16020, Egypt; 3Biochemistry Department, Faculty of Pharmacy, Helwan University, Ein Helwan, Cairo 11795, Egypt; 4Ministry of Health, Oman College of Health Sciences, Muscat 132, Oman; 5Medicinal Chemistry Department, Faculty of Pharmacy, Zagazig University, Zagazig 44519, Egypt

**Keywords:** sulfonamides, antioxidant, DPPH, 15-LOX, molecular docking

## Abstract

New antioxidant agents are urgently required to combat oxidative stress, which is linked to the emergence of serious diseases. In an effort to discover potent antioxidant agents, a novel series of 2-thiouracil-5-sulfonamides (**4**–**9**) were designed and synthesized. In line with this approach, our target new compounds were prepared from methyl ketone derivative **3**, which was used as a blocking unit for further synthesis of a novel series of chalcone derivatives **4a**–**d**, thiosemicarbazone derivatives **5a**–**d**, pyridine derivatives **6a**–**d** and **7a**–**d**, bromo acetyl derivative **8**, and thiazole derivatives **9a**–**d**. All compounds were evaluated as antioxidants against 2,2-diphenyl-1-picrylhydrazyl (DPPH), hydrogen peroxide (H_2_O_2_), lipid peroxidation, and 15-lipoxygenase (15-LOX) inhibition activity. Compounds **5c**, **6d**, **7d**, **9b**, **9c**, and **9d** demonstrated significant RSA in all three techniques in comparison with ascorbic acid and 15-LOX inhibitory effectiveness using quercetin as a standard. Molecular docking of compound **9b** endorsed its proper binding at the active site pocket of the human 15-LOX which explains its potent antioxidant activity in comparison with standard ascorbic acid.

## 1. Introduction

Lipoxygenases (LOXes) are nonheme iron-containing enzymes that regio- and stereo-specifically add oxygen to 1,4-polyunsaturated fatty acids to form the appropriate hydroperoxy derivatives [1,2]. Hydroperoxyeicosatetraenoic acid (HPETE) and hydroperoxyoctadecadienoic acid (HPODE), respectively, are the main lipid peroxidation products from arachidonic acid (AA) and linoleic acid (LA) [2]. These are easily transformed into the related hydroxy fatty acids hydroxy eicosatetraenoic acid (HETE) and hydroxy octadecadienoic acid (HODE). Plants, animals, and microorganisms all contain LOXes [3]. The various isozymes are known as 5-, 12-, or 15-LOXes because they can add a hydroperoxy moiety at carbons 5, 12, or 15 when arachidonic acid is the substrate. The 15-LOXes, LOX-1 and LOX-3, are lipoxygenases found in soybeans (Glycine max). Different mammalian LOX isoforms, including 5-LOX, 12-LOX, and 15-LOX, were discovered to be implicated in the etiology and development of numerous human disorders, which drew attention to the development of new pharmacological candidates that may function as potential lipoxygenase inhibitors [4]. The ability of the enzymes 15-LOX-1 and 15-LOX-2 to peroxide linoleic and arachidonic acids differently in animals has been established. It has been established that the aforementioned enzymes, as well as their metabolites hydroxy octadecadienoic acid (HODE), lipoxins, and eoxins, play a crucial role in the development of a number of diseases.

It’s interesting to note that research has shown that 15-LOX plays a factor in cardiovascular issues, including atherosclerosis, because it affects how low-density lipoproteins (LDL) are oxidized [5]. In addition, 15-LOX has been linked to the pathogenesis of neurological conditions like Alzheimer’s disease [6], the development of specific cancer cell types [7,8], and the production of pro-inflammatory and thrombotic mediators from the oxidation of arachidonic and linoleic acids [9,10]. As a result, 15-LOX is unquestionably a significant therapeutic target [11]. Therefore, LOXs are a viable target for the development of mechanism-based inhibitors and rational drug design for the management of inflammation, bronchial asthma, cancer, and autoimmune disorders.

Antioxidants act as a crucial line of defense against toxic effects generated by radicals by protecting the damages caused by free radicals. In the prevention and treatment of serious diseases like Alzheimer’s disease, atherosclerosis, cancer, diabetes, and stroke, antioxidants are beneficial [12]. The applications of antioxidants are widespread in numerous industries, including food, beverage, and cosmetics [13] etc. As more data is continuously acquired connecting the development of human diseases to oxidative stress, there has been an increase in interest in the application of antioxidants to medical treatment in recent years. Any biological system needs to maintain a delicate balance between the production of reactive oxygen and nitrogen species (ROS and RNS). ROS and RNS are produced regularly by either excessive oxidative stress or normal organ functions. Superoxide (O_2_^−^), hydrogen peroxide (H_2_O_2_), nitrogen oxide (NO), peroxynitrite (ONO_2_^−^), and hypochlorous acid (HOCl) are reactive species that, in excess, can have harmful effects and destroy tissue [14]. Organs use a variety of defense mechanisms, including endogenous and exogenous antioxidants, to protect themselves from the toxicity of excess ROS/RNS and maintain an oxido/redox balance [15].

Three crucial enzymes in these endogenous mechanisms are glutathione reductase (GSH), superoxide dismutase (SOD), and catalase. Exogenous antioxidants include substances that can scavenge free radicals, such as α-tocopherol, vitamin C, carotenoids, and polyphenols with a herbal origin. In the context of normal physiologic conditions, these enzymes and antioxidant substances balance the production and neutralization of free radicals. The use of dietary antioxidants, whether they are organic or synthetic, can boost defense against free radicals, enhance the quality of life by preventing numerous diseases, and significantly reduce the cost of health care delivery. In case of overproduction of free radicals in the body, they become inadequate or defective. Thus, the design and synthesis of novel antioxidant agents with high efficiency and low toxicity to hold this balance again is a growing research area in the field of medicinal chemistry [16,17].

Antioxidants can be divided into four main groups based on their mode of action [18,19]:(i)Chelators of metal ions involved in catalyzing lipid oxidation;(ii)Free radical scavengers;(iii)Lipoxygenase inactivators;(iv)Oxygen scavengers that react with oxygen in closed systems.

Most of the organic chemistry research globally focuses on heterocyclic chemistry. Pyrimidines, well-known heterocyclic compounds, occupy a special role in our life. This heterocyclic moiety is extremely important in biology and medicine. Numerous pyrimidine medicines have a range of therapeutic properties like anticancer such as 5-fluorouracil (5-FU), antiviral such as Idoxuridine and Trifluridine, anti-HIV such as Zidovudine and Stavudine, antibacterial such as Trimethoprim, Sulphamethiazine, and Sulphadiazine, antihypertensive such as Minoxidil and Prazosin, antithyroid such as Propylthiouracil, and antibiotic such as Bacimethrine [20]. Pyrimidine is a crucial component of nucleic acids, and it is used in pharmaceutics as a building block for the synthesis of anti-inflammatory, anti-hypertensive, antioxidant, anti-SARS [21,22,23], antiviral [24], anticancer [25], and antibacterial agents [26].

Literature is enriched with different pyrimidines; all have high activity as antioxidants [27,28,29,30,31]. As shown in Figure 1, pyrimidine derivatives **I** and **II** were reported to exhibit promising antioxidant activity [27,28]; thiazolopyrimidine derivatives **III** and **IV** combined with carbohydrazide, amino, and oxadiazol moieties possessed potential antioxidant activities [29]. 2-Hydrazinyl-4-(3-methoxyphenyl)-1-methyl-6-oxo-1,6-dihydropyrimidine-5-carb-onitrile **V** showed the most potent antioxidant activity by using scavenging of nitric oxide radical method and scavenging of hydrogen peroxide method [30]. Pyrimidine derivatives having sulfone moieties **VI** and **VII** were reported to have prominent antioxidant properties in both nitric oxide and DPPH methods [31] at a concentration of 100 μM. 4,6-Bisaryl-pyrimidin-2-amine derivative **VIII** showed good antioxidant activity as compared with ascorbic acid [32].

Additionally, the exceptional antiviral, anticancer, and antibacterial properties of the thiouracil derivatives make them of particular interest in medicinal chemistry [33,34,35]. 2-Thiouracils **IX**, **X**, and **XI** proved to have promising antioxidant activities compared to ascorbic acid [30,36]. Compound **XII** showed promising antioxidant activity with the IC_50_ of 0.6 mg/mL compared to gallic acid (IC_50_ = 0.0008 mg/mL) [37]. Compound **XIII** showed more promising antioxidant activity in comparison to standard butylated hydroxy toluene (Figure 2) [38].

The development of 15-LOX inhibitors for medicinal purposes is of great interest. Some heterocyclic substances significantly inhibit 15-LOX. Sulfonamides **XIV** (IC_50_ = 7 nM) [39]; **XV** (IC_50_ = 17 nM (LA), 50 nM (AA) [40,41], and **XVI** (IC_50_ = 10 μM) [39] showed potent inhibition of human 15-LOX. Moreover, thiourea **XVII** (IC_50_ rabbit enzyme = 2 nM), thiazole **XVIII** (IC_50_ = 0.096 μM), and isothiazolo derivative **XIX** (IC_50_ human = 0.12 μM) were found to work as a potent 15-LOX-1 inhibitors [39] (Figure 3).

The antioxidant and 15-LOX inhibitory activity shown by the above derivatives has drawn our interest in the development of new antioxidant agents. The synthesis and biological evaluation of new 2-thiouracil-5-sulfonamide derivatives incorporating other heterocyclic and non-heterocyclic derivatives are reported herein (Figure 4), which were tested for their potential as antioxidant agents against 2,2-diphenyl-1-picrylhydrazyl (DPPH), hydrogen peroxide (H_2_O_2_), lipid peroxidation and 15-lipoxygenase (15-LOX) inhibition activity. Future studies will discover the necessary antioxidant parameters that are most dependable in the design of 15-LOX inhibitors based on the in vitro antioxidant activity of the newly designed hybrids and their 15-LOX inhibitory activity. These derivatives’ potential modes of action and the structure-activity relationship (SAR) were also examined.

## 2. Results and Discussion

### 2.1. Chemistry

The literature review on the significance of 2-thiouracils, sulfonamides, and 2-thiouracil-5-sulfonamides in biological systems [42,43] inspired us to design and synthesize a new class of 5-substituted-2-thiouracil derivatives like the sulfonamide isosteres [44,45].

The synthetic route to prepare 2-thiouracil-5-sulfonamide derivatives (**2**–**7a**–**e**) is depicted in Figure 1 and Figure 2. The pyrimidine nucleus performs poorly in comparison to most frequent electrophilic substitution processes, including nitration, sulphonation, chlorination, etc. The -I and -M effects of the two nitrogen atoms are thought to be responsible for the relative inertness. The addition of groups that release electrons may enhance these reactions. In 2-thiouracil, the presence of OH and SH groups counteract the deactivation caused by the two nitrogen atoms. Consequently, 2-thiouracil undergoes chlorosulphonation [42,43]. The reaction of 2-thiouracil **1** with chlorosulphonic acid [44] at 120 °C yielded the targeted chlorosulfonyl analog **2**. Further, 2-thiouracil-5-sulfonyl chloride **2** was introduced in reaction with m-aminoacetophenone in absolute ethanol containing pyridine as an acid scavenger [44], giving the sulfonamide derivative **3**. Claisen-Schmidt condensation of **3** with some aromatic aldehydes (namely, 3-nitro benzaldehyde,3-chlorobenzaldehyde, 3-fluorobenzaldehyde and 3-methylbenzaldehyde) in 10% NaOH in ethanol, resulting in chalcones **4a**–**c**. Moreover, sulfonamide **3** also reacted with thiosemicarbazide, namely methyl, ethyl, 4-methoy phenyl and 4-flurophenyl thiosemicarbazides in absolute ethanol to provide the corresponding thiosemicarbazones **5a**–**c**, respectively (Figure 1).

Furthermore, 3-cyanopyridin-2-one or 2-amino-3-cyanopyridine derivatives **6a**–**d** and **7a**–**d**, respectively, were prepared from compound **3** via condensation with active methylene, namely ethyl cyanoacetate or malononitrile, the appropriate aromatic aldehydes and ammonium acetate [45]. The reaction of compound **3** with bromine in glacial acetic acid afforded the bromo derivative **8**, which was then utilized as a substrate for the synthesis of the thiazole derivatives **9a**–**d** by its reaction with several thiosemicarbazones (Figure 2). The structures of all the newly synthesized compounds (**4**–**9**) were completely consistent with the achieved spectral and elemental analysis data.

### 2.2. Molecular Modeling

Molecular docking was performed for the most active compound, **9b**, along with the redocking of protocatechuic acid and the substrate mimic that is cocrystallized in the 15-LOX-2 crystal (4RNE).

As previously detailed in the experimental section, protocatechuic acid interacts with the Ile676, Glu369 and Leu374, whereas the larger size of the substrate mimic assuming a U-shaped bioactive conformer that is further dipped into the active site by its two arms. The inner part of the substrate mimic overlapping with protocatechuic acid interacts with a structural water molecule that interacts directly with the iron cofactor. Interestingly, the Ile676 interacts with the Fe^+2^ ion on the opposite side of the water molecule. Therefore, the interaction between the substrate mimic and Ile676 is bridged by water and the Fe^+2^ ion (Figure 5).

By studying the docking results of the synthesized molecules **9b**, it reveals a binding mode that coincides with the substrate mimic (Figure 6). Herein, we detail the analysis for **9b** since it is the most active compound and scoring −9.8 Kcal/mol, which is close to the docking score of the co-crystallized substrate mimic of −10.9 Kcal/mol (Table 1). The bridging sulphonamide group of **9b** is located close to the structural water allowing for similar interactions to those observed by the co-crystallized substrate mimic. The central aromatic ring that is bonded to the sulphonamide group shows a π-π stacking with the His 378. The chlorophenyl moiety extends deep into the hydrophobic pocket, similar to the substrate mimic, but slightly shorter resulting in a slightly higher affinity score, i.e., −9.8 Kcal/mol. The choro substituent is positioned close to the carbonyl of the Ala 606, showing polar interaction, which might be attributed to the order of activity of **9b**, **9c**, and **9d**, which possess chloro, fluoro, and methyl substituents, respectively.

### 2.3. Biological Screening

#### 2.3.1. In-Vitro Antioxidant Activity

Results of screening of newly synthesized 2-thiouracil-5-Sulfonamide derivatives **3**–**9** as antioxidant molecules tested using 2,2-diphenyl-1-picrylhydrazyl (DPPH), hydrogen peroxide scavenging activity (H_2_O_2_), and lipid peroxidation assays (Figure 7). Ascorbic Acid (AA) was used as the antioxidant reference standard. The results of in vitro antioxidant activities were expressed as IC_50_ values (Table 2).

#### 2.3.2. DPPH Scavenging Activity

The antioxidant potential of all target compounds (**3**–**9**) was determined using DPPH radical scavenging assay in comparison with ascorbic acid (AA) as the control treatment.

The mechanism of antioxidants as DPPH radical scavengers is based on their hydrogen-donation ability as hydrogen atoms or an electron transfer from the compound to the DPPH to form DPPH-H. The results of DPPH reduction are presented in Table 2, Figure 7.

Generally, **3**–**9** exerted remarkable free radical scavenging efficacies in the range of 7.55 ± 1.70 μg/mL to 80.0 ± 0.70 μg/mL of IC_50_ against DPPH radical. Out of the 22 tested 2-thiouracil-5-Sulfonamide derivatives, twenty exhibited moderate to potent activity, which indicates their radical scavenging and their reducing activities.

The thiazole 2-thiouracils (**9a**–**d**) and 2- thiouracil thiosemicarbazones (**5a**–**d**) have potent antioxidant activities, while chalcones (**4a**–**d**) have displayed moderate RSA.

The most active compounds were **9b**, **9c**, **9d**, **5b**, and **5c** (IC_50_ = 11.9 ± 1.40, 10.0 ± 0.83, 7.55 ± 1.70, 14.00 ± 0.22, and 9.0 ± 1.10 μg/mL, respectively). They revealed potent RSA compared to ascorbic acid (IC50 = 12.80 ± 0.90 μg/mL).

Among the tested series of compounds, thiazoles present the highest activity. It was found that the presence of the pi-electron of excessive heterocycle thiazole enhances the antioxidant activity by increasing their electron donor capacity. The thiazole has no proton to donate to DPPH radical, but it is rich in electrons, and we proposed, due to its structure, that it donates electrons to stabilize the DPPH radical. Moreover, good radical scavenger properties of S atom C=S and free NH that act as a hydrogen donor in 2-thiouracil thiosemicarbazones showed potent DPPH RSA.

Compounds **9a**, **5a**, and **5d** (IC_50_ = 15.0 ± 1.25, 14.50 ± 0.72, and 14.72 ± 1.11 μg/mL, respectively) exhibited good RSA but lower than ascorbic acid. In addition, 3-cyanopyridin-2-one 2-thiouracils **6a**–**d** and 2-amino-3-cyanopyridine 2-thiouracils **7a**–**d** exhibited good activity in comparison to standard treatment AA. The lower DPPH RSA of chalcones (**4a**–**d**) proves the significant role of pi electrons of thiazole and NH_2_ of pyridine in antioxidant activity. All compounds (**3**–**9**) showed higher antioxidant RSA than their precursors, 2-thiouracil-5-Sulfonamide (**3**), thus indicating that thiazole and pyridine rings enhance the RSA of these compounds.

These data showed that compounds **6a**–**d** display much better antioxidant activity due to the introduction of pyridine rings. Moreover, it is obvious that the synergistic effect of the amino group and pyridine would enhance the antioxidant activity of 2-thiouracil derivatives. The results further confirm that the pyridine ring and amino group grafted into compounds **7a**–**d** contribute to the antioxidant action and consequently increase their antioxidant activity.

The antioxidant results of the DPPH assay are consistent with the conclusion that amino groups can enhance antioxidant activities [46]. The amino group in pyridine acts as an electron donor to quench free radicals by providing an electron, conceivably via an electron attack on the free radicals [47,48].

The SAR (Structure-Activity Relationship) study showed that the antioxidant activity of compounds tested by DPPH assay depends on the type of heterocyclic pharmacophore and also on substituents R/R^−^ on the aromatic ring of chalcone, thiazole, thiosemicarbazone and pyridine since antioxidant activity is related to electron or hydrogen donation capacity to DPPH**^.^** Radicals (Figure 8).

First, differences in the aromatic groups (thiazole, pyridine, and thiosemicarbazones) may be enhanced the antioxidant potency as the order of free radical scavenging activity (FRSA) was found to be: **9d** > **5c** > **7d** > **6d**.

Thiazoles **9** and thiosemicarbazones **5** have higher FRSA than amino pyridines **7** and pyridones **6**. For compounds **9d** and **5c**, their potent antioxidant activity is due to the presence of thiazole and semi-carbazones moiety [49,50]. While the introduction of pyridones gives lower antioxidant activity, as shown in pyridone derivatives **6**.

Second, concerning substitution patterns of chalcone, thiazole, thiosemicarbazone and pyridine, the order of antioxidant activity of compounds in descending order was found to be:

4-CH_3_ (**9d**) > 4-F(**9c**) > 4-Cl (**9b**)> 4-NO_2_ (**9a**).

It was also observed that the presence of electron-donating groups (-CH_3_ and -OCH_3_) led to an increase in antioxidant activity, while the presence of halogen atoms (Cl and F) on the benzene rings led to a decrease in oxidant activity.

The alkylated compounds exhibited more significant DPPH RSA than the corresponding halogenated compounds. 2-thiouracil derivative**s 5c** and **9d** display the most potent antioxidant activity, both having 4-CH_3_ or 4-OCH_3_ substituents on the phenyl ring, which is in accordance with the reported results. It was found that the best antioxidant capacity was found for the flavone substituted with a *p*-tolyl group on the thiazole ring [51].

During the DPPH assay, it was observed that the presence of electron-donating groups such as OCH_3_ and CH_3_ are more beneficial than mono chloro or fluoro-substituted phenyl ring and nitro compound, which may be due to +I and mesomeric effects [52]. These suggest that electron-donating groups on the aromatic ring induce antioxidant activity by the donation of electrons to the aromatic ring to activate it either by the resonance effect or the inductive effect [53].

This all indicates that the physicochemical properties of the designed molecule revealed an important role in the extent of its antioxidant activity. It is notable that the calculated c Log P for these derivatives is high (more lipophilic) compared to the standard treatment. In addition, these derivatives are cyclized heterocyclic analogs with fewer rotatable bonds that make them more favorable for cellular permeability compared to the standard treatment. It was found that the more electron donors substitutions on the aromatic side chain of the heterocyclic ring, the more antioxidant activity was observed.

#### 2.3.3. Hydrogen Peroxide Scavenging Activity

Hydrogen peroxide can inactivate a few enzymes directly, usually by oxidation of essential thiol (-SH) groups, as it is a weak oxidizing agent [54]. H_2_O_2_ can cross cell membranes rapidly, and it can react with Fe^+2^ and possibly Cu^+2^ inside the cell to form hydroxyl radicals. The latter causes severe damage to biological systems [55]. Therefore, scavenging H_2_O_2_ is very important for the protection of biological systems.

H_2_O_2_ assay is used to estimate the scavenging power of the target compounds (**4**–**9**) to H_2_O_2_. The results recorded in Table 2 showed that thiazole 2-thiouracils **9a**–**d** displayed higher H_2_O_2_ scavenging potential than the chalcone counterparts **4a**–**d** when compared to the ascorbic acid as a reference standard. Thiazole 2-thiouracil (**9d**) was the most potent H_2_O_2_ scavenger (IC_50_ = 15.0 μg/mL) with 1.8-fold that of ascorbic acid (IC_50_ = 23.0 μg g/mL) (Figure 5).

This signifies the role of electron-releasing group CH_3_ at the para-substituted group of benzene over thiazole for the inhibition of free radicals. 2-thiouracil thiosemicarbazones (**5a**–**d**) exhibited a moderate inhibitory effect in the hydrogen peroxide radical scavenging activity. Additionally, aminopyridines (**7a**–**d;** IC_50_ = 26.0 ± 1.28, 25.0 ± 1.35, 24.0 ± 1.40, and 23.9 ± 1.34 μg/mL, respectively) and pyridones (**6a**–**d;** IC_50_ = 31.0 ± 1.30, 30.0 ± 1.21, 29.0 ± 1.03 and 28.8 ± 1.05 μg/mL respectively) exhibited comparable H_2_O_2_ scavenging activity to ascorbic acid. The significantly lower antioxidant activity of chalcones (**4a**–**d**) confirms that chalcone moiety was not a favorable substitution over thiouracil ring for the antioxidant potential of the tested compounds against the H_2_O_2_ assay.

#### 2.3.4. Lipid Peroxidation Assay

Lipid peroxidation (LPO) caused by free radicals is supposed to be a primary mechanism of cell membrane destruction and cell damage [56].

The damage to lipids (by lipid peroxidation) has been reported to occur in three stages: initiation, propagation, and termination reactions. LPO may be initiated by radical species, which are sufficiently reactive to abstract a hydrogen atom from the unsaturated fatty acids. This is the starting point for the lipid radical chain propagation reaction. The propagation cycle is stopped by termination reactions (two radical species bond to form non-radical final products) which result in the destruction of free radicals.

New target compounds (**3**–**9**) were evaluated for inhibition of microsomal lipid peroxidation (LPO), where the ability of the compounds to scavenge free radicals was confirmed by microsomal lipid peroxidation inhibition in a liposome model system [57]. In this assay, lipid peroxidation is defined as the oxidative deterioration of polyunsaturated lipids [58], where the 50% inhibitory concentrations (IC_50_) were calculated (Table 2). The obtained results showed the ability of the tested compounds to exhibit significant antioxidant potential. Compounds **9a**–**d** exhibited pronounced antioxidant activity (IC_50_ = 22.0 ± 1.40, 21.0 ± 1.45, 20.5 ± 1.48, 20.0 ± 1.56, and 20.0 ± 1.56 μg/mL, respectively), which was higher than the standard ascorbic acid (IC_50_ = 36.0 ± 1.30 μg/mL) against lipid peroxidation. Meanwhile, the analogs **7a**–**d** (IC_50_ = 37.6 ± 1.2, 37.3 ± 1.3, 36.9 ± 1.72, and 36.4 ± 1.33 μg/mL, respectively) were nearly equiactive to ascorbic acid. Compounds **5a**–**d** have shown moderate inhibitory effects in the lipid peroxidation assay (Figure 7).

The results of the tested compounds (**4**–**9**) as antioxidants against LPO emphasized the important role of thiazole in the antioxidant activity in this assay regardless the substitution patterns were either 4-NO_2_ (**9a**) or 4-Cl (**9b**) or 4-F (**9c**) or4-CH_3_ (**9d**). Those derivatives were even superior to ascorbic acid in the antioxidant potential. But the contribution of pyridone and pyridine rings in controlling the activity of the generated hybrids couldn’t be ignored. So, the lower activity upon removal of thiazole, pyridone and pyridine rings for compounds (**9d**), (**7d**), and (**6d**) is highly expected according to the IC_50_ values reported for the compounds. This is because the antioxidant activity appeared to be (**9d**) > (**7d**) > (**6d**) to confirm the superiority of thiazole-thiouracil over both pyridone-thiouracil and pyridine-thiouracil and infer the role of thiazole in discriminating the potential antioxidant properties among potent compounds. This study also provides evidence that the presence of C=O, CN, or NH_2_ groups in **7d** and **6d** could extend its reaction with free radicals and terminate lipid peroxidation.

### 2.4. In Vitro 15-Lipoxygenase Inhibition Activity

All target compounds **3**–**9** were subjected to enzyme assay investigations against the Soybean 15-LOX enzyme. Results for the in vitro enzyme inhibition assays, displayed in Table 3, revealed that compounds **9a**, **9b**, **9c**, and **9d** exhibited potential 15-LOX inhibition activity when compared to quercetin (IC_50_ = 3.6 µM) as a reference inhibitor. Thiazoles **9b** and **9c**, in which the phenyl ring is substituted with Cl or F group, were the most potent compounds (IC_50_ = 1.80 ± 0.06 and 1.95 ± 0.06 µM, respectively) with 2.0- and 1.84-fold greater activity than that of quercetin respectively. Moreover, thiosemicarbazone **5c** (IC_50_ = 5.5 ± 0.02 µM), **5b** (IC_50_ = 5.7 ± 0.03 µM), **5a** (IC_50_ = 5.79 ± 0.01 µM), **5d** (IC_50_ = 5.9 ± 0.0 µM); pyridones **6d** (IC_50_ = 7.5 ± 0.04 µM), **6b** (IC_50_ = 7.6 ± 0.05 µM), **6c** (IC_50_ = 7.8 ± 0.02 µM), **6a** (IC_50_ = 7.85 ± 0.01µM); amino pyridine **7d** (IC_50_ = 6.6 ± 0.05 µM), **7b** (IC_50_ = 6.7 ± 0.01 µM), **7c** (IC_50_= 6.91 ± 0.02 µM), **7a** (IC_50_ = 6.95 ± 0.07 µM respectively), displayed good 15-LOX inhibitory activity but lower than quercetin. The results of the tested compounds as 15-LOX inhibitors emphasized the important role of thiazoles, thiosemicarbazone, pyridones and amino pyridine in this enzymatic assay. Those derivatives were superior to quercetin in 15-LOX inhibition. 15-LOX inhibition appeared to be (**9b** > **9c** > **9d** > **9a** > **5c** > **7d** > **6d**) to confirm the excel thiazole 2-thiouracil -5-sulfonamide over thiosemicarbazone and pyridine for antioxidant activity of the tested compounds against Soybean 15-LOX enzyme. It was also observed that halogenated derivatives showed significant 15-LOX inhibition activity. This might be due to the better fitting of the derivative into the catalytic pocket of the 15-LOX enzyme. In summary, compounds **5c**, **6d**, **7d**, **9a**, **9b**, **9c**, and **9d** exhibited significant RSA in all three methods in comparison with ascorbic acid and 15-LOX inhibition potency using quercetin as standard. This suggests an important influence of EDGs (CH_3_, OCH_3_) and halogens (Cl, F) in the benzene ring. Regarding heterocyclic pharmacophore, thiazole 2-thiouracil-5-sulfonamide showed higher RSA and 15-LOX inhibition potency than thiosemicarbazone and pyridine, and these observations should be regarded in the future design of LOX inhibitors.

## 3. Materials and Methods

### 3.1. Instruments

Using the Electro-thermal IA 9100 equipment from Shimadzu, Kyoto, Japan, all melting points were calculated and were uncorrected. On a PerkinElmer 1650 spectrophotometer, FT-IR spectra were taken using potassium bromide pellets (USA). Utilizing a Varian Mercury (300 MHz and 75 MHz, respectively) spectrometer (Varian, Crawley, UK), 1H-NMR and 13C-NMR spectra were acquired in DMSO-d6, and chemical shifts were presented as ppm from TMS as an internal reference. The 70 eV EI Ms-QP 1000 EX was used to record mass spectra. The results of the microanalyses, which were carried out by the Organic Microanalysis Unit, Faculty of Science, Cairo University, Cairo, Egypt, using the Vario and Elementar apparatus, were within the calculated values’ acceptable range (0.40). Silica gel 60 was subjected to column chromatography at Merck, Darmstadt, Germany (particle size 0.06–0.20 mm).

### 3.2. Chemistry

#### 3.2.1. 2-Thiouracil-5-sulphonyl Chloride (**2**): Prepared as in Literature [44]

A mixture of 2-thiouracil 1 (12.5 g, 0.055 mol) and chlorosulphonic acid (51 mL, 0.055 mol) was heated at 120 °C for 8 h. The reaction mixture was cooled and poured on a mixture of ice and acetic acid 1:1; the precipitate was filtered off, dried under suction, and used as crude for subsequent work. Yield: 63%, m.p. 230 °C; as reported [44].

#### 3.2.2. N-(3-Acetylphenyl)-4-oxo-2-thioxo-1,2,3,4-tetrahydropyrimidine-5-sulfonamide (**3**): Prepared as in Literature [59]

A mixture of pyrimidine-5-sulfonyl chloride **2** (1.13 g, 0.005 mol), 3-aminoacetophenone (0.005 mol), and pyridine (0.4 mL, 0.005 mol) was refluxed in 25 mL absolute ethanol for 16–20 h, then cooled, filtered off, dried, and recrystallized from DMF/water. Yield: 78%; m.p.: 318–320 °C; as reported [59].

#### 3.2.3. General Procedure for the Preparation of Compounds (**4a**–**d**)

A mixture of 3 (1.01 g, 0.003 mol) and the suitable aldehyde (0.003 mol) in 50 mL ethanolic sodium hydroxide solution was agitated at room temperature for 24 h before being refluxed for one hour, cooled, and then poured into ice-cold water. The precipitate that formed after neutralization with dil HCl was filtered out, washed with water, dried, and recrystallized from DMF/water.


**N-{3-[(2E)-3-(3-nitrophenyl)prop-2-enoyl]phenyl}-4-oxo-2-thioxo-1,2,3,4-tetrahydro pyrimidine-5-sulfonamide: (4a).**


Yield 67%, m.p. 240–242 °C. IR (KBr) v max (cm^−1^): 3236 (NH), 3170 (CH-Ar); 1670, 1695, (2C=O), 1553, 1348 (NO_2_), 1145, 1333 (SO_2_). ^1^H-NMR (300 MHz, DMSO-d_6_) 6.6, 6.8 (d, 2H, J = 6.4 Hz, CH=CH), 7.1–7.5 (m, 8H, Ar-H), 8.2 (s, 1H, pyrimidine), 10.1, 10.3, 11.1 (3 s, 3H, 3NH, D_2_O exchangeable). ^13^C-NMR (300 MHz, DMSO-d_6_) 103 (C-5 pyrimidine), 113.1–149.9 (SP^2^ carbon atoms), 150 (C-6 pyrimidine), 168.3, 187 (C=O), 183 (C=S); MS (EI): *m/z* 458 [M+] (4.2%); Anal. Calcd. for C_19_H_14_N_4_O_6_S_2_ (458.46): C, 49.82; H, 3.08; N, 12.22; Found: C, 49.81; H, 3.09; N, 12.27.


**N-{3-[(2E)-3-(3-chlorophenyl)prop-2-enoyl]phenyl}-4-oxo-2-thioxo-1,2,3,4-tetra hydropyrimidine-5-sulfonamide: (4b).**


Yield 69%, m.p. 283–285 °C. IR (KBr) v max (cm^−1^): 3242 (NH), 3183 (CH-Ar); 1678, 1696, (2C=O), 1144, 1337 (SO_2_). ^1^H-NMR (300 MHz, DMSO-d_6_) 6.5, 6.7 (d, 2H, J = 6.4 Hz, CH=CH), 7.2–7.5 (m, 8H, Ar–H), 8.1 (s, 1H, pyrimidine), 10.0, 10.3, 11.1(3 s, 3H, 3NH, D_2_O exchangeable). ^13^C-NMR (300 MHz, DMSO-d_6_) 103.3 (C-5 pyrimidine), 113.5–148.9 (SP^2^ carbon atoms), 150.2 (C-6 pyrimidine), 168.5, 187.2 (C=O), 183.2 (C=S); MS (EI): *m/z* 447 [M+] (10.3%), 449 (M+2, 3.6%); Anal. Calcd. for C_19_H_14_ClN_3_O_4_S_2_ (447.91): C, 50.95; H, 3.15; N, 9.38; Found: C, 50.88; H, 3.17; N, 9.47.


**N-{3-[(2E)-3-(3-fluorophenyl)prop-2-enoyl]phenyl}-4-oxo-2-thioxo-1,2,3,4-tetra hydropyrimidine-5-sulfonamide: (4c).**


Yield 65%, m.p. 270–272 °C. IR (KBr) v max (cm^−1^): 3259 (NH), 3193 (CH-Ar); 1675, 1690, (2C=O), 1140, 1336 (SO_2_). ^1^H-NMR (300 MHz, DMSO-d_6_) 6.7, 6.7 (d, 2H, J = 7.4 Hz, CH=CH), 7.1–7.5 (m, 8H, Ar–H), 8.1 (s, 1H, pyrimidine), 10.1, 10.3, 11.1 (3 s, 3H, 3NH, D_2_O exchangeable). ^13^C-NMR (300 MHz, DMSO-d_6_) 91.88 (C-5 pyrimidine), 111.41–147.20 (SP^2^ carbon atoms), 152.72 (C-6 pyrimidine), 68.12, 175.72 (C=O), 183.07 (C=S); MS (EI): *m/z* 431 [M+] (21.3%); Anal. Calcd. for C_19_H_14_FN_3_O_4_S_2_ (431.46): C, 52.89; H, 3.27; N, 9.74; Found: C, 52.74; H, 3.31.; N, 9.73.


**N-{3-[(2E)-3-(3-methylphenyl)prop-2-enoyl]phenyl}-4-oxo-2-thioxo-1,2,3,4-tetra hydropyrimidine-5-sulfonamide: (4d).**


Yield 61%, m.p. 287–289 °C. IR (KBr) v max (cm^−1^): 3243 (NH), 3187 (CH-Ar); 1671, 1692, (2C=O), 1143, 1339 (SO_2_). ^1^H-NMR (300 MHz, DMSO-d_6_) 2.5 (s, 3H, CH_3_), 6.6, 6.8 (d, 2H, J = 7.4 Hz, CH=CH), 7.1–7.7 (m, 8H, Ar–H), 8.1 (s, 1H, pyrimidine), 11.1, 11.4, 11.6 (3 s, 3H, 3NH, D_2_O exchangeable). ^13^C-NMR (300 MHz, DMSO-d_6_), 21.2 (CH_3_), 103.5 (C-5 pyrimidine), 113.7–148.7 (SP^2^ carbon atoms), 150.5 (C-6 pyrimidine), 168.6, 187.7 (C=O), 183.3 (C=S); MS (EI): *m/z* 427 [M+] (19.9%); Anal. Calcd. for C_20_H_17_N_3_O_4_S_2_ (427.49): C, 56.19; H, 4.01; N, 9.83; Found: C, 56.23; H, 4.09; N, 9.76.

#### 3.2.4. General Procedure for the Preparation of Compounds (**5a**–**d**)

A mixture of 3 (1.1 g,0.003 mol) and the suitable substituted thiosemicarbazide (0.003 mol) was refluxed in 30 mL absolute ethanol for 12–15 h, followed by cooling, filtering out, drying, and recrystallization from DMF/water.


**N-{3-[(1E)-N-(methylcarbamothionyl)ethanhydrazonoyl]phenyl}-4-oxo-2-thioxo-1,2,3,4-tetrahydropyrimidine-5-sulfonamide (5a).**


Yield 74%, m.p. 305–307 °C. IR (KBr) v max (cm^−1^): 3285 (NH), 3197 (CH-Ar) 2980, (CH-sp^3^), 1687 (C=O), 1271 (2C=S), 1141, 1335 (SO_2_). ^1^H-NMR (300 MHz, DMSO-d_6_) 2.0, 2.5 (s, 6H, 2CH_3_), 7.0–7.5 (m, 4H, Ar–H), 8.5 (s, 1H, pyrimidine), 9.5, 10.0, 10.5, 11.4, 11.5 (5 s, 5H, 5NH, D_2_O exchangeable). ^13^C-NMR (300 MHz, DMSO-d_6_) 18.3 and 33.7 (2 CH_3_), 103.1 (C-5 pyrimidine), 117.1–146.4 (SP^2^ carbon atoms), 150.0 (C-6 pyrimidine), 155.5 (C=N), 168.1 (C=O), 183.0, 184.1 (C=S); MS (EI): *m/z* 412 [M+] (4.4%); Anal. Calcd. for C_14_H_16_N_6_O_3_S_3_ (412.51): C, 40.76; H, 3.91; N, 20.37; Found: C, 40.75; H, 3.81; N, 20.42.


**N-{3-[(1E)-N-(ethylcarbamothionyl)ethanhydrazonoyl]phenyl}-4-oxo-2-thioxo-1,2,3,4-tetrahydropyrimidine-5-sulfonamide (5b).**


Yield 74%, m.p. 310–312 °C. IR (KBr) v max (cm^−1^): 3283 (NH), 3179 (CH-Ar) 2967, (CH-sp^3^), 1680 (C=O), 1271 (2C=S), 1140, 1336 (SO_2_). ^1^H-NMR (300 MHz, DMSO-d_6_) 2.1 (s, 3H, CH_3_), 2.3(q, 2H, CH_2_), 1.9(t, 3H, CH_3_), 7.2–7.4 (m, 4H, Ar–H), 8.1 (s, 1H, pyrimidine), 10.0, 10.1, 11.0, 11.1, 11.2 (5 s, 5H, 5NH, D_2_O exchangeable). ^13^C-NMR (300 MHz, DMSO-d_6_) 15.5, 18.3 (CH_3_), 47.4 (CH_2_), 103.3 (C-5 pyrimidine), 117.3–146.6 (SP^2^ carbon atoms), 150.1 (C-6 pyrimidine), 155.5 (C=N), 168.6 (C=O), 183.1,184.4 (C=S); MS (EI): *m/z* 426 [M+] (14.3%); Anal. Calcd. for C_15_H_18_N_6_O_3_S_3_ (426.53): C, 42.24; H, 4.25; N, 19.70C; Found: C, 42.35; H, 4.35; N, 19.71.


**N-{3-[(1E)-N-(4-methoxycarbamothionyl)ethanhydrazonoyl]phenyl}-4-oxo-2-thioxo-1,2,3,4-tetrahydropyrimidine-5-sulfonamides: (5c).**


Yield 78%, m.p. 307–309 °C. IR (KBr) v max (cm^−1^): 3293 (NH), 3165 (CH-Ar) 2950, (CH-sp^3^), 1680 (C=O), 1271 (2C=S), 1141, 1337 (SO_2_). ^1^H-NMR (300 MHz, DMSO-d_6_) 2.5 (s, 3H, CH_3_), 3.5 (s, 3H, OCH_3_), 7–7.5 (m, 4H, Ar–H), 8.1 (s, 1H, pyrimidine), 9.0, 9.5, 10.0, 11.2, 11.5 (5 s, 5H, 5NH, D_2_O exchangeable). ^13^C-NMR (300 MHz, DMSO-d_6_) 15.2 (CH_3_), 56.0 (OCH_3_), 103.4 (C-5 pyrimidine), 117.5–146.5 (SP^2^ carbon atoms), 150.2 (C-6 pyrimidine), 155.5 (C=N), 168.7 (C=O), 183.3, 184.5 (C=S); MS (EI): *m/z* 504 [M+] (14.3%); Anal. Calcd. for C_20_H_20_N_6_O_4_S_3_ (504.6): C, 47.60; H, 3.99; N, 16.65; Found: C, 47.58; H, 3.86; N, 16.71.


**N-{3-[(1E)-N-(4-fluorocarbamothionyl)ethanhydrazonoyl]phenyl}-4-oxo-2-thioxo-1,2,3,4-tetrahydropyrimidine-5-sulfonamides: (5d).**


Yield 77%, m.p. 313–315 °C. IR (KBr) v max (cm^−1^): 3294 (NH), 3177 (CH-Ar) 2989, (CH-sp3), 1686 (C=O), 1271 (2C=S), 1141, 1337 (SO_2_). ^1^H-NMR (300 MHz, DMSO-d_6_) 2.3 (s, 3H, CH_3_), 7.1–7.4 (m, 8H, Ar–H), 8.1 (s, 1H, pyrimidine), 10.1, 10.2, 11.0, 11.0, 11.1 (5 s, 5H, 5NH, D_2_O exchangeable).^13^C-NMR (300 MHz, DMSO-d_6_) 15.23 (CH_3_), 93.88 (C-5 pyrimidine), 103.65–142.47 (SP^2^ carbon atoms), 164.04 (C-6 pyrimidine), 158.04 (C=N), 168.02 (C=O), 172.03, 180.11 (C=S); MS (EI): *m/z* 492 [M+] (20.6%); Anal. Calcd. for C_19_H_17_FN_6_O_3_S_3_ (492.57): C, 46.33; H, 3.48; N, 17.06.; Found: C, 46.43; H, 3.46; N, 17.15.

#### 3.2.5. General Procedure for the Preparation of Compounds (6a–d)

A mixture of 3 (1.1 gm, 0.003 mol), the necessary aldehydes (0.003 mol), ammonium acetate (1.89 gm, 8 mol), and ethyl cyanoacetate (0.35 gm, 0.003 mol) in 50 mL absolute ethanol was refluxed for 8–10 h. The reaction mixture was concentrated to equal its half volume, filtered off, the filtrate was placed into ice/water, and the precipitate was filtered off, dried, and recrystallized from DMF/water.


**N-{3-[5-cyano-4-(3-nitrophenyl)-6-oxo-1,6-dihydropyridin-2-yl]phenyl}-4-oxo-2-thioxo-1,2,3,4-tetrahydropyrimidine-5-sulfonamide (6a).**


Yield 65%, m.p. 266–268 °C. IR (KBr) v max (cm^−1^): 3267 (NH), 3186 (CH-Ar), 2220 (CN), 2971, (CH-sp^3^), 1680, 1694 (2C=O), 1271 (C=S), 1141, 1337 (SO_2_). ^1^H-NMR (300 MHz, DMSO-d_6_) 7.2–7.5 (m, 8H, Ar–H), 7.9 (1H, s, pyridone), 8.1 (s, 1H, pyrimidine), 9.5, 10.0, 10.1, 11.0 (4 s, 4H, 4NH, D_2_O exchangeable).^13^C-NMR (300 MHz, DMSO-d_6_) 103.1 (C-5 pyrimidine), 117.1 (CN), 112.1–169.4 (SP^2^ carbon atoms), 150.1 (C-6 pyrimidine), 162.9, 168.3 (2C=O), 178.2(C=S); MS (EI): *m/z* 522 [M+] (10.3%); Anal. Calcd. for C_22_H_14_N_6_O_6_S_2_ (522.51): C, 50.57; H, 2.70; N, 16.08.; Found: C, 50.68; H, 2.80; N, 16.11.


**N-{3-[5-cyano-4-(3-chlorophenyl)-6-oxo-1,6-dihydropyridin-2-yl]phenyl}-4-oxo-2-thioxo-1,2,3,4-tetrahydropyrimidine-5-sulfonamide: (6b).**


Yield 65%, m.p. 272–274 °C. IR (KBr) v max (cm^−1^): 3283 (NH), 3183 (CH-Ar) 2973 (CH-sp3), 2226(CN), 1682, 1697(2C=O), 1271 (C=S), 1143, 1332 (SO_2_). ^1^H-NMR (300 MHz, DMSO-d_6_) 7.1–7.6 (m, 8H, Ar–H), 7.9 (1H, s, pyridone), 8.1 (s, 1H, pyrimidine), 9.6, 10.1, 10.2, 11.1 (4 s, 4H, 4NH, D_2_O exchangeable). ^13^C-NMR (300 MHz, DMSO-d_6_) 95.78 (C-5 pyrimidine), 115.28 (CN), 98.48–169.48 (SP^2^ carbon atoms), 159.66 (C-6 pyrimidine), 162.42, 168.51 (2C=O), 175.72 (C=S); MS (EI): *m/z* 511 [M+] (15.3%), 513 (M+2, 4.4%).; Anal. Calcd. for C_22_H_14_ClN_5_O_4_S_2_ (511.96): C, 51.61; H, 2.76; N, 13.68.; Found: C, 51.69; H, 2.80; N, 13.57.


**N-{3-[5-cyano-4-(3-fluorophenyl)-6-oxo-1,6-dihydropyridin-2-yl]phenyl}-4-oxo-2-thioxo-1,2,3,4-tetrahydropyrimidine-5-sulfonamide (6c).**


Yield 64%, m.p. 287–289 °C. IR (KBr) v max (cm^−1^): 3267 (NH), 3191 (CH-Ar) 2987 (CH-sp3), 2222 (CN), 1686, 1695 (2C=O), 1271 (C=S), 1141, 1330 (SO_2_). ^1^H-NMR (300 MHz, DMSO-d_6_) 7.2–7.6 (m, 8H, Ar–H), 7.9 (1H, s, pyridone), 8.1 (s, 1H, pyrimidine), 9.8, 10.1, 10.2, 11.1 (4 s, 4H, 4NH, D_2_O exchangeable). ^13^C-NMR (300 MHz, DMSO-d_6_) 103.6 (C-5 pyrimidine), 117.1 (CN), 100.3–169.6 (SP^2^ carbon atoms), 150.2 (C-6 pyrimidine), 162.3, 168.3 (2C=O), 178.7 (C=S); MS (EI): *m/z* 495 [M+] (19.7%); Anal. Calcd. for C_22_H_14_FN_5_O_4_S_2_ (495.50): C, 53.33; H, 2.85; N, 14.13; Found: C, 53.39; H, 2.70; N, 14.09.


**N-{3-[5-cyano-4-(3-methylphenyl)-6-oxo-1,6-dihydropyridin-2-yl]phenyl}-4-oxo-2-thioxo-1,2,3,4-tetrahydropyrimidine-5-sulfonamide: (6d).**


Yield 61%, m.p. 282–284 °C. IR (KBr) v max (cm^−1^): 3287 (NH), 3187 (CH-Ar) 2985 (CH-sp3), 2222 (CN), 1684, 1696 (2C=O), 1271 (C=S), 1143, 1332 (SO_2_). ^1^H-NMR (300 MHz, DMSO-d_6_) 3.5 (3H, s, CH_3_), 7.3–7.6 (m, 8H, Ar–H), 7.9 (1H, s, pyridone), 8.1 (s, 1H, pyrimidine), 9.7, 10.0, 10.1, 11.1 (4 s, 4H, 4NH, D_2_O exchangeable). ^13^C-NMR (300 MHz, DMSO-d_6_) 21.1 (CH_3_), 103.5 (C-5 pyrimidine), 117.1 (CN), 100.2–169.9 (SP^2^ carbon atoms), 150.2 (C-6 pyrimidine), 162.3, 168.3 (2C=O), 178.3 (C=S); MS (EI): *m/z* 491 [M+] (13.3%); Anal. Calcd. for C_23_H_17_N_5_O_4_S_2_ (491.54): C, 56.20; H, 3.49; N, 14.25; Found: C, 56.36; H, 3.50; N, 14.31.

#### 3.2.6. General Procedure for the Preparation of Compounds (**7a**–**d**)

A mixture of 3 (1.1 gm, 0.003 mol), the necessary aldehydes (0.003 mol), ammonium acetate (1.89 gm, 8 mol), and malononitrile (0.2 gm, 0.003 mol) in 50 mL absolute ethanol was refluxed for 8–10 h. The reaction mixture was concentrated to equal its half volume. After being filtered off, the filtrate was placed into ice/water, and the precipitate was filtered off, dried, and recrystallized from DMF/water.


**N-{3-[6-amino-5-cyano-4-(3-nitrophenyl)pyridine-2-yl]phenyl}-4-oxo-2-thioxo-1,2,3,4-tetrahydropyrimidine-5-sulfonamides: (7a).**


Yield 62%, m.p. 251–253 °C. IR (KBr) v max (cm^−1^): 3292 (NH), 3183 (CH-Ar) 2983 (CH-sp3), 2225 (CN), 1684 (C=O), 1350, 1550 (NO_2_), 1271 (C=S), 1141, 1337 (SO_2_). ^1^H-NMR (300 MHz, DMSO-d_6_) 7.2–7.5 (m, 9H, Ar–H), 8.1 (s, 1H, pyrimidine), 9.1, 10.1, 10.2, 11.1 (4 s, 5H, 4NH, D_2_O exchangeable). ^13^C-NMR (300 MHz, DMSO-d_6_) 93.0 (C-5 pyrimidine), 114.0 (CN), 90.5–164.7 (SP2 carbon atoms),152.0 (C-6 pyrimidine), 168.5 (C=O), 178.0 (C=S); MS (EI): *m/z* 521 [M+] (10.4%)); Anal. Calcd. for C_22_H_15_N_7_O_5_S_22_ (521.52): C, 50.67; H, 2.90; N,18.80; Found: C, 50.68; H, 2.86; N, 18.78.


**N-{3-[6-amino-5-cyano-4-(3-chlorophenyl)pyridine-2-yl]phenyl}-4-oxo-2-thioxo-1,2,3,4-tetrahydropyrimidine-5-sulfonamides: (7b).**


Yield 67%, m.p. 257–259 °C. IR (KBr) v max (cm^−1^): 3267 (NH), 3186 (CH-Ar) 2987 (CH-sp3), 2222 (CN), 1680 (C=O), 1271 (C=S), 1140, 1338 (SO_2_). ^1^H-NMR (300 MHz, DMSO-d_6_) 7.2–7.6 (m, 9H, Ar–H), 8.1 (s, 1H, pyrimidine), 9.2, 10, 10.2, 11.1 (4 s, 5H, 4NH, D_2_O exchangeable). ^13^C-NMR (300 MHz, DMSO-d_6_) 103.3 (C-5 pyrimidine), 117.7 (CN), 100.1–165.5 (SP^2^ carbon atoms), 150.5 (C-6 pyrimidine), 168.6 (C=O), 178.7 (C=S); MS (EI): *m/z* 510 [M+] (14.7%) 512 (M+2, 4.9%); Anal. Calcd. for C_22_H_15_ClN_6_O_3_S_2_ (510.97): C, 51.71; H, 2.96; N, 16.45; Found: C, 51.68; H, 2.88; N, 16.58.


**N-{3-[6-amino-5-cyano-4-(3-fluorophenyl)pyridine-2-yl]phenyl}-4-oxo-2-thioxo-1,2,3,4-tetrahydropyrimidine-5-sulfonamides: (7c).**


Yield 72%, m.p. 268–270 °C. IR (KBr) v max (cm^−1^): 3283 (NH), 3181 (CH-Ar) 2972 (CH-sp3), 2222 (CN), 1679 (C=O), 1271 (C=S), 1140, 1338 (SO_2_). ^1^H-NMR (300 MHz, DMSO-d_6_) 7.2–7.7 (m, 9H, Ar–H), 8.1 (s, 1H, pyrimidine), 9.0, 10, 10.3, 11.6 (4 s, 5H, 4NH, D_2_O exchangeable). ^13^C-NMR (300 MHz, DMSO-d_6_) 103.3 (C-5 pyrimidine), 117.1 (CN), 100.3–165.6 (SP^2^ carbon atoms), 150.1 (C-6 pyrimidine),168.5 (C=O), 178.5 (C=S); MS (EI): *m/z* 494 [M+] (10.6%); Anal. Calcd. for C_22_H_15_FN_6_O_3_S_2_ (494.52): C, 53.43; H, 3.06; N, 16.99.; Found: C, 53.49; H, 3.10; N, 16.82.


**N-{3-[6-amino-5-cyano-4-(3-methylphenyl)pyridine-2-yl]phenyl}-4-oxo-2-thioxo-1,2,3,4-tetrahydropyrimidine-5-sulfonamides: (7d).**


Yield 70%, m.p. 264–266 °C. IR (KBr) v max (cm^−1^): 3281 (NH), 3172 (CH-Ar) 2969 (CH-sp^3^), 2222 (CN), 1680 (C=O), 1271 (C=S), 1141, 1338 (SO_2_). ^1^H-NMR (300 MHz, DMSO-d_6_) 3.6 (s,3H, CH_3_), 7.1–7.7 (m, 9H, Ar–H), 8.1 (s, 1H, pyrimidine), 9.1, 10, 10.2, 11.5 (4 s, 5H, 4NH, D_2_O exchangeable). ^13^C-NMR (300 MHz, DMSO-d_6_) 21.1 (CH_3_),103.5 (C-5 pyrimidine), 117.1 (CN), 100.2–165.5 (SP^2^ carbon atoms), 150.1 (C-6 pyrimidine), 168.6 (C=O), 178.5 (C=S); MS (EI): *m/z* 490 [M+] (14.7%); Anal. Calcd. for C_23_H_18_N_6_O_3_S_2_ (490.55): C, 56.31; H, 3.70; N, 17.13; Found: C, 56.40; H, 3.80; N, 17.28.


**N-[3-(bromoacetyl)phenyl]-4-oxo-2-thioxo-1,2,3,4-tetrahydropyrimidine-5-sulfonamide (8).**


A mixture of 3 (1.13 g, 0.005 mol) and bromine (0.005 mol) in 30 mL glacial acetic acid was stirred at room temperature for 48 h, then filtered. The filtrate was neutralized with ammonia, and the precipitate was collected, filtered, dried, and recrystallized from DMF/water.

Yield 80%, m.p. 305–307 °C. IR (KBr) v max (cm^−1^): 3295 (NH), 3197 (CH-Ar) 2958 (CH-sp3), 1680, 1685 (2C=O), 1275 (C=S), 1140, 1330 (SO_2_). ^1^H-NMR (300 MHz, DMSO-d_6_) 3.8 (2H, s, CH_2_), 6.5–7.3 (m, 4H, Ar–H), 7.5 (s, 1H, pyrimidine), 10, 11, 11.5 (3 s, 3H, 3NH, D_2_O exchangeable). ^13^C-NMR (300 MHz, DMSO-d_6_) 43.2 (CH_2_), 96.2 (C-5 pyrimidine), 112.7–144.1 (SP^2^ carbon atoms), 150.8 (C-6 pyrimidine), 168.3, 189.7 (C=O), 175.3 (C=S); MS (EI): *m/z* 404 [M+] (15.3%), 406 (M+2, 15%); Anal. Calcd. for C_12_H_10_BrN_3_O_4_S_2_ (404.25): C, 35.65; H, 2.49; N, 10.39; Found: C, 35.79; H, 2.55; N, 10.32.

#### 3.2.7. General Procedure for the Preparation of Compounds (**9a**–**d**)

A mixture of 8 (1.1 g, 0.003 mol) and the appropriate thiosemicarbazone derivatives (0.003 mol) in 40 mL absolute ethanol was refluxed for 15–18 h, then the reaction mixture was cooled, and the formed solid was filtered off, dried and recrystallized from DMF/water.


**N-(3-{2-[(2E)-2(3-nitrobenzylidene)hydrazine]-1,3-thiazol-4-yl}phenyl)-4-oxo-thioxo-1,2,3,4-tetrahydropyrimidine-5-sulfonamides: (9a).**


Yield 64%, m.p.: 316–318 °C. IR (KBr) v max (cm^−1^): 3293 (NH), 3190 (CH-Ar) 2963 (CH-sp3), 1683 (C=O), 1275 (C=S), 1351, 1555 (NO_2_), 1140, 1335 (SO_2_). ^1^H-NMR (300 MHz, DMSO-d_6_) 6.7 (s, 1H, CH=N), 6.8–7.4 (m, 8H, Ar–H), 7.7 (1H, s, thiazole), 8.1 (s, 1H, pyrimidine), 6.5, 9.5, 10.7, 10.8 (4 s, 4H, 4NH, D_2_O exchangeable). ^13^C-NMR (300 MHz, DMSO-d_6_) 103.0 (C-5 pyrimidine), 113.5–158.2 (SP^2^ carbon atoms), 150.0 (C-6 pyrimidine),155.3 (C=N), 168.3 (C=O), 178.0 (C=S); MS (EI): *m/z* 529.57 [M+] (8.7%); Anal. Calcd. for C_20_H_15_N_7_O_5_S_3_ (529.57): C, 45.36; H, 2.85; N, 18.; Found: C, 45.70; H, 2.79; N, 18.62.


**N-(3-{2-[(2E)-2(3-chlorobenzylidene)hydrazine]-1,3-thiazol-4-yl}phenyl)-4-oxo-thioxo-1,2,3,4-tetrahydropyrimidine-5-sulfonamides: (9b).**


Yield 60%, m.p.: 320–322 °C. IR (KBr) v max (cm^−1^): 3287 (NH), 3195 (CH-Ar) 2978 (CH-sp^3^), 1680 (C=O), 1270 (C=S), 1141, 1338 (SO_2_). ^1^H-NMR (300 MHz, DMSO-d_6_) 6.8 (s, 1H, CH=N), 6.9–7.7 (m, 8H, Ar–H), 7.7 (1H, s, thiazole), 8.2 (s, 1H, pyrimidine), 6.4, 11.0, 11.4, 11.56 (4 s, 4H, 4NH, D_2_O exchangeable). ^13^C-NMR (300 MHz, DMSO-d_6_) 93.9 (C-5 pyrimidine), 104.8–159.7 (SP^2^ carbon atoms), 152.0 (C-6 pyrimidine), 154.8 (C=N), 164.0 (C=O), 170.0 (C=S); MS (EI): *m/z* 518 [M+] (9.3%), 520 (M+2, 3.1%); Anal. Calcd. for C_20_H_15_ClN_6_O_3_S_3_ (519.01): C, 46.28; H, 2.91; N, 16.19; Found: C, 46.31; H, 2.95; N, 16.28.


**N-(3-{2-[(2E)-2(3-fluorobenzylidene)hydrazine]-1,3-thiazol-4-yl}phenyl)-4-oxo-thioxo-1,2,3,4-tetrahydropyrimidine-5-sulfonamides: (9c).**


Yield 63%, m.p.: 317–319 °C. IR (KBr) v max (cm^−1^): 3285 (NH), 3195 (CH-Ar) 2985 (CH-sp3), 1689 (C=O), 1270 (C=S), 1145, 1340 (SO_2_). ^1^H-NMR (300 MHz, DMSO-d6) 6.5 (s, 1H, CH=N), 6.6–7.5 (m, 8H, Ar–H), 7.7 (1H, s, thiazole), 8.2 (s, 1H, pyrimidine), 6.7, 9.1, 10.3, 10.8 (4 s, 4H, 4NH, D_2_O exchangeable). ^13^C-NMR (300 MHz, DMSO-d_6_), 103.5 (C-5 pyrimidine), 113.8–159.8 (SP^2^ carbon atoms), 150.0 (C-6 pyrimidine), 155.2 (C=N), 168.8 (C=O), 178.8 (C=S); MS (EI): *m/z* 502 [M+] (10.3%); Anal. Calcd. for C_20_H_15_FN_6_O_3_S_3_ (502.56): C, 47.80; H, 3.01; N, 16; Found: C, 47.79; H, 3.18; N, 16.83.


**N-(3-{2-[(2E)-2(3-methylbenzylidene)hydrazine]-1,3-thiazol-4-yl}phenyl)-4-oxo-thioxo-1,2,3,4-tetrahydropyrimidine-5-sulfonamides: (9d).**


Yield 63%, m.p.: 326–328 °C. IR (KBr) v max (cm^−1^): 3283 (NH), 3192 (CH-Ar) 2983 (CH-sp^3^), 1686 (C=O), 1270 (C=S), 1143, 1340 (SO_2_). ^1^H-NMR (300 MHz, DMSO-d_6_) 3.5 (s, 3H, CH_3_), 6.6 (s, 1H, CH=N), 6.7–7.4 (m, 8H, Ar–H), 7.7 (1H, s, thiazole), 8.1(s, 1H, pyrimidine), 6.8, 9.3, 10.5, 10.7 (4 s, 4H, 4NH, D_2_O exchangeable). ^13^C-NMR (300 MHz, DMSO-d_6_) 20.1 (CH_3_), 103.5 (C-5 pyrimidine), 113.5–159.8 (SP2 carbon atoms), 150.0 (C-6 pyrimidine), 155.7 (C=N), 168.6 (C=O), 178.8 (C=S); MS (EI): *m/z* 498 [M+] (10.8%); Anal. Calcd. for C_21_H_18_N_6_O_3_S_3_ (498.60): C, 50.59; H, 3.64; N, 16.86.; Found: C, 50.60; H, 3.58; N, 16.88.

### 3.3. Biological Evaluation

#### 3.3.1. In-Vitro Assays for Biological Antioxidant Activity

##### Chemicals

The analytical grade chemicals necessary for all experiments were purchased from Sigma-Aldrich Chemicals Co., in St. Louis, MO, USA.

#### 3.3.2. DPPH Scavenging Method

The DPPH scavenging activity of all novel synthesized compounds (**3**–**9**) was measured as previously described [60] (see Appendix A).

#### 3.3.3. Scavenging of Hydrogen Peroxide

The hydrogen peroxide (H_2_O_2_) scavenging activity of all novel synthesized compounds **3**–**9** was measured as previously reported [61] (see Appendix A).

#### 3.3.4. Lipid Peroxidation Assay

The Lipid peroxidation activity of all novel synthesized compounds **3**–**9** was measured as previously reported [62] (see Appendix A).

### 3.4. In-Vitro Lipoxygenase Inhibition Activity

All target thiouracil analogs **3**–**9** were further tested for 15-LOX inhibitory activity using Cayman’s Lipoxygenase Inhibitor Screening Assay Kit (Catalog No. 760700, Cayman Chemical, USA.

90 µL of 15-LOX was pipetted into a 96-well plate quickly. The test chemical was then dissolved in DMSO in 10 µL portions at concentrations of 2.5 µM, 5.0 µM, and 10 µM and added to each well. Arachidonic acid, a 110 µL substrate, was added to start the reaction, and the plate was shaken for at least five minutes. In order to interrupt enzyme catalysis and advance the reaction, 100 µL of chromogen was added to each well and made in accordance with the manufacturer’s instructions. In blank wells, 100 µL of assay buffer (0.1 M Tris-HCl, pH 7.4) was utilized. The positive control and 100% beginning activity were Quercetin and DMSO, respectively. The solution’s absorbance was determined at λ 490–500 nm. The percentage inhibition was calculated according to the following equation:% inhibition = [(IA − A_inhibitor sample_)/IA] × 100(1)
where (IA) is the 100% initial activity and (A_inhibitor sample)_ is the absorbance of the test sample. A dose-response curve was plotted between % inhibition and the drug concentration. The non-linear dose-response curve was used for calculating drug concentration showing 50% enzyme inhibition.

### 3.5. Molecular Modeling

Molecular docking was performed for the most active compound **9b** to elucidate the mode of binding and explain their activity. The available crystal structure of the human 15-LOX-2 allows for a reliable prediction of intermolecular interactions [63]. Furthermore, previous crystallography studies on solutions of quercetin and LOX protein from soybeans revealed that protocatechuic acid (3,4-dihydroxybenzoic acid) positions the catechol OH groups towards Ile857 (forming an H-bond) and the carboxyl group between Gln514 and Trp519. The H-bond between Ile857 and the OH of protocatechuic acid [64] is possible, as this is actually the carboxylic terminus of the protein which interacts with the iron cofactor (see Appendix A).

## 4. Conclusions

The present study introduced the synthesis of a novel series of 2-thiouracil-5-sulphonamides **4**–**9**, and their spectral and elemental analyses proved chemical structures. The antioxidant activity of all synthesized compounds was screened against 2,2-diphenyl-1-picrylhydrazyl (DPPH), hydrogen peroxide (H_2_O_2_), lipid peroxidation and 15-lipoxygenase (15-LOX) inhibition activity. The most active compounds were **9b**, **9c**, **9d**, **5b**, and **5c** (IC_50_ = 11.9 ± 1.40, 10.0 ± 0.83, 7.55 ± 1.70, 14.70 ± 0.22, and 9.0 ± 1.10 μg/mL, respectively). They revealed more potent RSA than ascorbic acid (IC_50_ = 12.80 ± 0.90 μg/mL) against DPPH radical. The SAR (Structure-Activity Relationship) study showed that the order of free radical scavenging activity (FRSA) was found to be: (**9d** > **5c** > **7d** > **6d**). It was also observed that the presence of electron-donating groups (-CH_3_, -OCH_3_) led to an increase in antioxidant activity, while the presence of halogen atoms (Cl, F) on the benzene rings led to a decrease in oxidant activity. Chalcone **5c** and thiazole derivative **9d** show the most potent antioxidant activity, both having 4-CH_3/_or 4-OCH_3_ substituents on the phenyl ring, which is in accordance with the reported results. Thiazole (**9d**) was the most potent H_2_O_2_ scavenger (IC_50_ = 15.0 μg/mL) with 1.8 folds that of ascorbic acid (IC_50_ = 23.0 μg g/mL). Compounds **9a**–**d** exhibited pronounced antioxidant activity (IC_50_ = 22.0 ± 1.40, 21.0 ± 1.45, 20.5 ± 1.48, 20.0 ± 1.56, and 20.0 ± 1.56μg/mL, respectively), which was higher than the standard ascorbic acid (IC_50_ = 36.0 ± 1.30 μg/mL) against lipid peroxidation. Compounds **9b**, **9c**, and **9d** displayed potential 15-LOX inhibition activity when compared to quercetin (IC_50_ = 3.34 µM) as a reference inhibitor. Thiazoles **9b** and **9c**, in which the phenyl ring is substituted with a Cl or F group, were the most potent compounds (IC_50_ = 1.80 ± 0.06 and 1.95 ±0.06 µM, respectively) with 1.85 and 1.71 folds that of quercetin respectively. Moreover, thiosemicarbazone **5c** (IC_50_ = 5.5 ± 0.02 µM), pyridones **6d** (IC_50_ = 7.5 ± 0.04 µM), amino pyridine **7d** (IC_50_ = 6.6 ± 0.05 µM, respectively) displayed good 15-LOX inhibitory activity but lower than quercetin. In summary, compounds **5c**, **6d**, **7d**, **9b**, **9c**, and **9d** showed significant RSA in all three methods in comparison with ascorbic acid and 15-LOX inhibition potency using quercetin as standard. This suggests an important influence of EDGs (CH_3_, OCH_3_) and halogens (Cl, F) in the benzene ring. Regarding heterocyclic pharmacophore, thiazole showed higher RSA and 15-LOX inhibition potency than thiosemicarbazone and pyridine, and these observations should be regarded in the future on the designed LOX inhibitors. Molecular docking for the most active compound **9b** was carried out in order to understand the possible binding mode with the lipoxygenase enzyme. The binding pose of compound **9b** was perfectly correlated with the biological data and served to justify the observed antioxidant effect. In conclusion, the obtained results suggest that these potent compounds may serve as lead candidates for 15-LOX inhibitors. Furthermore, the designed thiouracil hybrid scaffold is an interesting antioxidant pharmacophore and is considered a novel lead scaffold for any future optimization.

## Data Availability

Not applicable.

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
