# Peer review of "Novel 2-Thiouracil-5-Sulfonamide Derivatives: Design, Synthesis, Molecular Docking, and Biological Evaluation as Antioxidants with 15-LOX Inhibition"

_molecules, 2023, doi:10.3390/molecules28041925_

Round 1

Reviewer 2 Report

The manuscript about target new compounds were prepared from methyl ketone derivative 3 which was used as a blocking unit for further synthesis of a novel series of chalcone derivatives 4a-d, thiosemicarbazone derivatives 5a-d, pyridine derivatives 6a-d and 7a-d, bromo acetyl derivative 8 and thiazole derivatives 9a-d.

I have read at great length the work. I found this article informative in regards to background information.

Thus, the main objective of this research is obvious and well elaborated.

The results in tables and figures are well presented, so I have no objection to their presentation. The results are adequately discussed and compared with other works. The conclusion confirms the obtained results and the references inserted in the main document show a good connection between this investigation and other works.

The material and methods, as well as results and discussion part are understandable. The article is good from grammatical and structural point of view, and for my perspective is acceptable for publication in this journal.

The manuscript appears to have sufficient scientific quality and may be of interest to the readers of the Journal.

The scientific value of the manuscript is good (although not outstanding)

The manuscript brings some new information. 

 Minor errors:

Abstract: first two sentences need to rewrite

Introduction

-          Need to add objective of the paper

Materials and methods: no statistical analysis but in general good.

Results : good but some figures resolution not good.

Discussion: good

References:

-          The abbreviations of some journals in References are incorrect

-          - References not on format of journal. 

Reviewer 3 Report

"Novel 2-Thiouracil-5-Sulfonamide Derivatives: design, synthe- 2 sis, molecular docking, and biological evaluation as antioxidant 3 with 15-LOX inhibition"  manuscript submitted is of good quality,  the article cannot be accepted as such in the present form. i reccomend for a revision.

 however i have few queries and suggestion to the authors

1. the formula of hydrogen peroxide is not proper throughout the manuscript and in figures, the authors should take take in providing the subscript and superscript for the chemical formula usage.

2. the word " showed" is seen through out the manuscript, preplace it either by exhibited or displays.

3. the Substitution of R in figure 1 is confusing, please provide some other alternative so that it can be easily understood by readers. 

4. the authors have mentioned " complicated diseaces " the terminolgy look little vague. is the authors consider thet there are simple dieseases as per your usage?

5. throughout the manuscript, regarding the N-H bond in figures, there doessnt seem any bond between them. provide the proper bonding as per the software used.

6. figure 3: provide the formula for n-pentyl.

7. Scheme 2: The reagents are not provided with the IUPAC momenclature, common usage names should be avoided and the abbreviations are not uniform.

8.  Figure 7: provide proper subscript and superscript in x and y axis (hydrogen peroxide and IC 50)

9. Figure 8: Clarity to be enhanced.

10.  "3.2. Chemistry"  line no 452. what does the sub-heading mean?

11. materials and methods can be sent to supplemntary section, some of the charecterisation data can be also sent to supplementary data.

12. References: an uniform format of the journal name to be provided. in some cases, there is a capiltalization and in some there are small letters.  
